# Online Decision-Making in General Combinatorial Spaces

**Arun Rajkumar**     **Shivani Agarwal**

Department of Computer Science and Automation
Indian Institute of Science, Bangalore 560012, India
`{arun_r,shivani}@csa.iisc.ernet.in`

## Abstract

We study *online combinatorial decision problems*, where one must make sequential decisions in some combinatorial space without knowing in advance the cost of decisions on each trial; the goal is to minimize the total regret over some sequence of trials relative to the best fixed decision in hindsight. Such problems have been studied mostly in settings where decisions are represented by Boolean vectors and costs are linear in this representation. Here we study a general setting where costs may be linear in any suitable low-dimensional vector representation of elements of the decision space. We give a general algorithm for such problems that we call *low-dimensional online mirror descent* (LDOMD); the algorithm generalizes both the Component Hedge algorithm of Koolen et al. (2010), and a recent algorithm of Suehiro et al. (2012). Our study offers a unification and generalization of previous work, and emphasizes the role of the convex polytope arising from the vector representation of the decision space; while Boolean representations lead to 0-1 polytopes, more general vector representations lead to more general polytopes. We study several examples of both types of polytopes. Finally, we demonstrate the benefit of having a general framework for such problems via an application to an online transportation problem; the associated transportation polytopes generalize the Birkhoff polytope of doubly stochastic matrices, and the resulting algorithm generalizes the PermELearn algorithm of Helmbold and Warmuth (2009).

## 1   Introduction

In an online combinatorial decision problem, the decision space is a set of combinatorial structures, such as subsets, trees, paths, permutations, etc. On each trial, one selects a combinatorial structure from the decision space, and incurs a loss; the goal is to minimize the regret over some sequence of trials relative to the best fixed structure in hindsight. Such problems have been studied extensively in the last several years, primarily in the setting where the combinatorial structures are represented by Boolean vectors, and costs are linear in this representation; this includes online learning of paths, permutations, and various other specific combinatorial structures [16, 17, 12], as well as the Component Hedge algorithm of Koolen et al. [14] which generalizes many of these previous studies. More recently, Suehiro et al. [15] considered a setting where the combinatorial structures of interest are represented by the vertices of the base polytope of a submodular function, and costs are linear in this representation; this includes as special cases several of the Boolean examples considered earlier, as well as new settings such as learning permutations with certain position-based losses (see also [2]).

In this work, we consider a general form of the online combinatorial decision problem, where costs can be linear in any suitable low-dimensional vector representation of the combinatorial structures of interest. This encompasses representations as Boolean vectors and vertices of submodular base polytopes as special cases, but also includes many other settings. We give a general algorithm for

such problems that we call *low-dimensional online mirror descent* (LDOMD); the algorithm generalizes both the Component Hedge algorithm of Koolen et al. for Boolean representations [14], and the algorithm of Suehiro et al. for submodular polytope vertex representations [15].[1] As we show, in many settings of interest, the regret bounds for LDOMD are better than what can be obtained with other algorithms for online decision problems, such as the Hedge algorithm of Freund and Schapire [10] and the Follow the Perturbed Leader algorithm of Kalai and Vempala [13].

We start with some preliminaries and background in Section 2, and describe the LDOMD algorithm and its analysis in Section 3. Our study emphasizes the role of the convex polytope arising from the vector representation of the decision space; we study several examples of such polytopes, including matroid polytopes, polytopes associated with submodular functions, and permutation polytopes in Sections 4–6, respectively. Section 7 applies our framework to an online transportation problem.

## 2  Preliminaries and Background

**Notation.** For $n \in \mathbb{Z}_+$, we will denote $[n] = \{1, \ldots, n\}$. For a vector $z \in \mathbb{R}^d$, we will denote by $\|z\|_1, \|z\|_2$, and $\|z\|_\infty$ the standard $L_1$, $L_2$, and $L_\infty$ norms of $z$, respectively. For a set $\mathcal{Z} \subseteq \mathbb{R}^d$, we will denote by $\mathrm{conv}(\mathcal{Z})$ the convex hull of $\mathcal{Z}$, and by $\mathrm{int}(\mathcal{Z})$ the interior of $\mathcal{Z}$. For a closed convex set $\mathcal{K} \subseteq \mathbb{R}^d$ and Legendre function $F : \mathcal{K} \to \mathbb{R}$,[2] we will denote by $B_F : \mathcal{K} \times \mathrm{int}(\mathcal{K}) \to \mathbb{R}_+$ the Bregman divergence associated with $F$, defined as $B_F(x, x') = F(x) - F(x') - \nabla F(x') \cdot (x - x')$, and by $F^* : \nabla F(\mathrm{int}(\mathcal{K})) \to \mathbb{R}$ the Fenchel conjugate of $F$, defined as $F^*(u) = \sup_{x \in \mathcal{K}} (x \cdot u - F(x))$.

**Problem Setup.** Let $\mathcal{C}$ be a (finite but large) set of combinatorial structures. Let $\phi : \mathcal{C} \to \mathbb{R}^d$ be some injective mapping that maps each $c \in \mathcal{C}$ to a unique vector $\phi(c) \in \mathbb{R}^d$ (so that $|\phi(\mathcal{C})| = |\mathcal{C}|$). We will generally assume $d \ll |\mathcal{C}|$ (e.g. $d = \mathrm{poly}\log(|\mathcal{C}|)$). The online combinatorial decision-making problem we consider can be described as follows: On each trial $t$, one makes a decision in $\mathcal{C}$ by selecting a structure $c^t \in \mathcal{C}$, and receives a loss vector $\ell^t \in [0,1]^d$; the loss incurred is given by $\phi(c^t) \cdot \ell^t$ (see Figure 1). The goal is to minimize the *regret* relative to the single best structure in $\mathcal{C}$ in hindsight; specifically, the regret of an algorithm $\mathcal{A}$ that selects $c^t \in \mathcal{C}$ on trial $t$ over $T$ trials is defined as

---

Online Combinatorial Decision-Making

**Inputs:**
   Finite set of combinatorial structures $\mathcal{C}$
   Mapping $\phi : \mathcal{C} \to \mathbb{R}^d$

**For** $t = 1 \ldots T$:
   – Predict $c^t \in \mathcal{C}$
   – Receive loss vector $\ell^t \in [0,1]^d$
   – Incur loss $\phi(c^t) \cdot \ell^t$

---

Figure 1: Online decision-making in a general combinatorial space.

$$R_T[\mathcal{A}] = \sum_{t=1}^{T} \phi(c^t) \cdot \ell^t - \min_{c \in \mathcal{C}} \sum_{t=1}^{T} \phi(c) \cdot \ell^t .$$

In particular, we would like to design algorithms whose worst-case regret (over all possible loss sequences) is sublinear in $T$ (and also has as good a dependence as possible on other relevant problem parameters). From standard results, it follows that for any deterministic algorithm, there is always a loss sequence that forces the regret to be linear in $T$; as is common in the online learning literature, we will therefore consider randomized algorithms that maintain a probability distribution $p^t$ over $\mathcal{C}$ from which $c^t$ is randomly drawn, and consider bounding the *expected* regret of such algorithms.

**Online Mirror Descent (OMD).** Recall that online mirror descent (OMD) is a general algorithmic framework for *online convex optimization* problems, where on each trial $t$, one selects a point $x^t$ in some convex set $\Omega \subseteq \mathbb{R}^n$, receives a convex cost function $f_t : \Omega \to \mathbb{R}$, and incurs a loss $f_t(x^t)$; the goal is to minimize the regret relative to the best single point in $\Omega$ in hindsight. The OMD algorithm makes use of a Legendre function $F : \mathcal{K} \to \mathbb{R}$ defined on a closed convex set $\mathcal{K} \supseteq \Omega$, and effectively performs a form of projected gradient descent in the dual space of $\mathrm{int}(\mathcal{K})$ under $F$, the projections being in terms of the Bregman divergence $B_F$ associated with $F$. See Appendix A.1 for an outline of OMD and its regret bound for the special case of *online linear optimization*, where costs $f_t$ are linear (so that $f_t(x) = \ell^t \cdot x$ for some $\ell^t \in \mathbb{R}^n$), which will be relevant to our study.

**Hedge/Naïve OMD.** The Hedge algorithm proposed by Freund and Schapire [10] is widely used for online decision problems in general. The algorithm maintains a probability distribution over the decision space, and can be viewed as an instantiation of the OMD framework, with $\Omega$ (and $\mathcal{K}$) the probability simplex over the decision space, linear costs $f_t$ (since one works with expected losses), and $F$ the negative entropy. When applied to online combinatorial decision problems in a naïve manner, the Hedge algorithm requires maintaining a probability distribution over the combinatorial decision space $\mathcal{C}$, which in many cases can be computationally prohibitive (see Appendix A.2 for an outline of the algorithm, which we also refer to as Naïve OMD). The following bound on the expected regret of the Hedge/Naïve OMD algorithm is well known:

**Theorem 1** (Regret bound for Hedge/Naïve OMD). *Let $\phi(c) \cdot \ell^t \in [a,b]\ \forall c \in \mathcal{C}, t \in [T]$. Then setting $\eta^* = \frac{2}{(b-a)} \sqrt{\frac{2 \ln |\mathcal{C}|}{T}}$ gives*

$$\mathbf{E}\Big[R_T\big[\operatorname{Hedge}(\eta^*)\big]\Big] \ \leq \ (b-a)\sqrt{\frac{T \ln |\mathcal{C}|}{2}}\,.$$

**Follow the Perturbed Leader (FPL).** Another widely used algorithm for online decision problems is the Follow the Perturbed Leader (FPL) algorithm proposed by Kalai and Vempala [13] (see Appendix A.3 for an outline of the algorithm). Note that in the combinatorial setting, FPL requires the solution to a combinatorial optimization problem on each trial, which may or may not be efficiently solvable depending on the form of the mapping $\phi$. The following bound on the expected regret of the FPL algorithm is well known:

**Theorem 2** (Regret bound for FPL). *Let $\|\phi(c) - \phi(c')\|_1 \leq D_1$, $\|\ell^t\|_1 \leq G_1$, and $|\phi(c) \cdot \ell^t| \leq B$ $\forall c, c' \in \mathcal{C}, t \in [T]$. Then setting $\eta^* = \sqrt{\frac{D_1}{BG_1 T}}$ gives*

$$\mathbf{E}\Big[R_T\big[\operatorname{FPL}(\eta^*)\big]\Big] \ \leq \ 2\sqrt{D_1 B G_1 T}\,.$$

**Polytopes.** Recall that a set $\mathcal{S} \subset \mathbb{R}^d$ is a *polytope* if there exist a finite number of points $x_1, \ldots, x_n \in \mathbb{R}^d$ such that $\mathcal{S} = \operatorname{conv}(\{x_1, \ldots, x_n\})$. Any polytope $\mathcal{S} \subset \mathbb{R}^d$ has a unique *minimal* set of points $x_1', \ldots, x_m' \in \mathbb{R}^d$ such that $\mathcal{S} = \operatorname{conv}(\{x_1', \ldots, x_m'\})$; these points are called the *vertices* of $\mathcal{S}$. A polytope $\mathcal{S} \subset \mathbb{R}^d$ is said to be a *0-1 polytope* if all its vertices lie in the Boolean hypercube $\{0,1\}^d$.

As we shall see, in our study of online combinatorial decision problems as above, the polytope $\operatorname{conv}(\phi(\mathcal{C})) \subset \mathbb{R}^d$ will play a central role. Clearly, if $\phi(\mathcal{C}) \subseteq \{0,1\}^d$, then $\operatorname{conv}(\phi(\mathcal{C}))$ is a 0-1 polytope; in general, however, $\operatorname{conv}(\phi(\mathcal{C}))$ can be any polytope in $\mathbb{R}^d$.

## 3   Low-Dimensional Online Mirror Descent (LDOMD)

We describe the Low-Dimensional OMD (LDOMD) algorithm in Figure 2. The algorithm maintains a point $x^t$ in the polytope $\operatorname{conv}(\phi(\mathcal{C}))$. It makes use of a Legendre function $F : \mathcal{K} \to \mathbb{R}$ defined on a closed convex set $\mathcal{K} \supseteq \operatorname{conv}(\phi(\mathcal{C}))$, and effectively performs OMD in a $d$-dimensional space rather than in a $|\mathcal{C}|$-dimensional space as in the case of Hedge/Naïve OMD. Note that an efficient implementation of LDOMD requires two operations to be performed efficiently: (a) given a point $x^t \in \operatorname{conv}(\phi(\mathcal{C}))$, one needs to be able to efficiently find a 'decomposition' of $x^t$ into a convex combination of a small number of points in $\phi(\mathcal{C})$ (this yields a distribution $p^t \in \Delta_{\mathcal{C}}$ that satisfies $\mathbf{E}_{c \sim p^t}[\phi(c)] = x^t$ and also has small support, allowing efficient sampling); and (b) given a point $\widetilde{x}^{t+1} \in \mathcal{K}$, one needs to be able to efficiently find a 'projection' of $\widetilde{x}^{t+1}$ onto $\operatorname{conv}(\phi(\mathcal{C}))$ in terms of the Bregman divergence $B_F$. The following regret bound for LDOMD follows directly from the standard OMD regret bound (see Theorem 4 in Appendix A.1):

**Theorem 3** (Regret bound for LDOMD). *Let $B_F(\phi(c), x^1) \leq D^2\ \forall c \in \mathcal{C}$. Let $\|\cdot\|$ be any norm in $\mathbb{R}^d$ such that $\|\ell^t\| \leq G\ \forall t \in [T]$, and such that the restriction of $F$ to $\operatorname{conv}(\phi(\mathcal{C}))$ is $\alpha$-strongly convex w.r.t. $\|\cdot\|_*$, the dual norm of $\|\cdot\|$. Then setting $\eta^* = \frac{D}{G}\sqrt{\frac{2\alpha}{T}}$ gives*

$$\mathbf{E}\Big[R_T\big[\operatorname{LDOMD}(\eta^*)\big]\Big] \ \leq \ DG\sqrt{\frac{2T}{\alpha}}\,.$$

As we shall see below, the LDOMD algorithm generalizes both the Component Hedge algorithm of Koolen et al. [14], which applies to settings where $\phi(\mathcal{C}) \subseteq \{0,1\}^d$ (Section 3.1), and the recent algorithm of Suehiro et al. [15], which applies to settings where $\operatorname{conv}(\phi(\mathcal{C}))$ is the base polytope associated with a submodular function (Section 5).

---

**Algorithm** Low-Dimensional OMD (LDOMD) for Online Combinatorial Decision-Making

---

**Inputs:**
    Finite set of combinatorial structures $\mathcal{C}$
    Mapping $\phi : \mathcal{C} \to \mathbb{R}^d$

**Parameters:**
    $\eta > 0$
    Closed convex set $\mathcal{K} \supseteq \mathrm{conv}(\phi(\mathcal{C}))$, Legendre function $F : \mathcal{K} \to \mathbb{R}$

**Initialize:**
    $x^1 = \mathrm{argmin}_{x \in \mathrm{conv}(\phi(\mathcal{C}))} F(x)$   (or $x^1 =$ any other point in $\mathrm{conv}(\phi(\mathcal{C}))$)

**For** $t = 1 \ldots T$:
  – Let $p^t$ be any distribution over $\mathcal{C}$ such that $\mathbf{E}_{c \sim p^t}[\phi(c)] = x^t$   [Decomposition step]
  – Randomly draw $c^t \sim p^t$
  – Receive loss vector $\ell^t \in [0,1]^d$
  – Incur loss $\phi(c^t) \cdot \ell^t$
  – Update:
      $\widetilde{x}^{t+1} \leftarrow \nabla F^*(\nabla F(x^t) - \eta \ell^t)$
      $x^{t+1} \leftarrow \mathrm{argmin}_{x \in \mathrm{conv}(\phi(\mathcal{C}))} B_F(x, \widetilde{x}^{t+1})$   [Bregman projection step]

---

Figure 2: The LDOMD algorithm.

### 3.1 LDOMD with 0-1 Polytopes

Consider first a setting where each $c \in \mathcal{C}$ is represented as a Boolean vector, so that $\phi(\mathcal{C}) \subseteq \{0,1\}^d$. In this case $\mathrm{conv}(\phi(\mathcal{C}))$ is a 0-1 polytope. This is the setting commonly studied under the term 'online combinatorial learning' [14, 8, 3]. In analyzing this setting, one generally introduces an additional problem parameter, namely an upper bound $m$ on the 'size' of each Boolean vector $\phi(c)$. Specifically, let us assume $\|\phi(c)\|_1 \leq m \ \forall c \in \mathcal{C}$ for some $m \in [d]$.

Under the above assumption, it is easy to verify that applying Theorems 1 and 2 gives

$$\mathbf{E}\Big[R_T\big[\,\mathrm{Hedge}(\eta^*)\,\big]\Big] = O\Big(m\sqrt{Tm\ln(\tfrac{d}{m})}\Big); \qquad \mathbf{E}\Big[R_T\big[\,\mathrm{FPL}(\eta^*)\,\big]\Big] = O(m\sqrt{Td}).$$

For the LDOMD algorithm, since $\mathrm{conv}(\phi(\mathcal{C})) \subseteq [0,1]^d \subset \mathbb{R}_+^d$, it is common to take $\mathcal{K} = \mathbb{R}_+^d$ and to let $F : \mathcal{K} \to \mathbb{R}$ be the unnormalized negative entropy, defined as $F(x) = \sum_{i=1}^d x_i \ln x_i - \sum_{i=1}^d x_i$, which leads to a multiplicative update algorithm; the resulting algorithm was termed Component Hedge in [14]. For the above choice of $F$, it is easy to see that $B_F(\phi(c), x^1) \leq m\ln(\tfrac{d}{m}) \ \forall c \in \mathcal{C}$; moreover, $\|\ell^t\|_\infty \leq 1 \ \forall t$, and the restriction of $F$ on $\mathrm{conv}(\phi(\mathcal{C}))$ is $(\tfrac{1}{m})$-strongly convex w.r.t. $\|\cdot\|_1$. Therefore, applying Theorem 3 with appropriate $\eta^*$, one gets

$$\mathbf{E}\Big[R_T\big[\,\mathrm{LDOMD}(\eta^*)\,\big]\Big] = O\Big(m\sqrt{T\ln(\tfrac{d}{m})}\Big).$$

Thus, when $\phi(\mathcal{C}) \subseteq \{0,1\}^d$, the LDOMD algorithm with the above choice of $F$ gives a better regret bound than both Hedge/Naïve OMD and FPL; in fact the performance of LDOMD in this setting is essentially optimal, as one can easily show a matching lower bound [3].

Below we will see how several online combinatorial decision problems studied in the literature can be recovered under the above framework (e.g. see [16, 17, 12, 14, 8]); in many of these cases, both decomposition and unnormalized relative entropy projection steps in LDOMD can be performed efficiently (in $\mathrm{poly}(d)$ time) (e.g. see [14]). As a warm-up, consider the following simple example:

**Example 1** ($m$-sets with element-based losses)**.** *Here $\mathcal{C}$ contains all size-$m$ subsets of a ground set of $d$ elements: $\mathcal{C} = \{S \subseteq [d] \,|\, |S| = m\}$. On each trial $t$, one selects a subset $S^t \in \mathcal{C}$ and receives a loss vector $\ell^t \in [0,1]^d$, with $\ell_i^t$ specifying the loss for including element $i \in [d]$; the loss for the subset $S^t$ is given by $\sum_{i \in S^t} \ell_i^t$. Here it is natural to define a mapping $\phi : \mathcal{C} \to \{0,1\}^d$ that maps each $S \in \mathcal{C}$ to its characteristic vector, defined as $\phi_i(S) = \mathbf{1}(i \in S) \ \forall i \in [d]$; the loss incurred on predicting $S^t \in \mathcal{C}$ is then simply $\phi(S^t) \cdot \ell^t$. Thus $\phi(\mathcal{C}) = \{x \in \{0,1\}^d \,|\, \|x\|_1 = m\}$, and $\mathrm{conv}(\phi(\mathcal{C})) = \{x \in [0,1]^d \,|\, \|x\|_1 = m\}$. LDOMD with unnormalized negative entropy as above has a regret bound of $O\big(m\sqrt{T\ln(\tfrac{d}{m})}\big)$. It can be shown that both decomposition and unnormalized relative entropy projection steps take $O(d^2)$ time [17, 14].*

### 3.2 LDOMD with General Polytopes

Now consider a general setting where $\phi : \mathcal{C} \rightarrow \mathbb{R}^d$, and $\mathrm{conv}(\phi(\mathcal{C})) \subset \mathbb{R}^d$ is an arbitrary polytope. Let us assume again $\|\phi(c)\|_1 \leq m \ \forall c \in \mathcal{C}$ for some $m > 0$.

Again, it is easy to verify that applying Theorems 1 and 2 gives

$$\mathbf{E}\Big[R_T\big[\,\mathrm{Hedge}(\eta^*)\,\big]\Big] \; = \; O(m\sqrt{T \ln |\mathcal{C}|}) ; \qquad \mathbf{E}\Big[R_T\big[\,\mathrm{FPL}(\eta^*)\,\big]\Big] \; = \; O(m\sqrt{Td}) \,.$$

For the LDOMD algorithm, we consider two cases:

**Case 1:** $\phi(\mathcal{C}) \subset \mathbb{R}_+^d$. Here one can again take $\mathcal{K} = \mathbb{R}_+^d$ and let $F : \mathcal{K} \rightarrow \mathbb{R}$ be the unnormalized negative entropy. In this case, one gets $B_F(\phi(c), x^1) \leq m \ln(d) + m \ \forall c \in \mathcal{C}$ if $m < d$, and $B_F(\phi(c), x^1) \leq m \ln(m) + d \ \forall c \in \mathcal{C}$ if $m \geq d$. As before, $\|\ell^t\|_\infty \leq 1 \ \forall t$, and the restriction of $F$ on $\mathrm{conv}(\phi(\mathcal{C}))$ is $(\frac{1}{m})$-strongly convex w.r.t. $\|\cdot\|_1$, so applying Theorem 3 for appropriate $\eta^*$ gives

$$\mathbf{E}\Big[R_T\big[\,\mathrm{LDOMD}(\eta^*)\,\big]\Big] \; = \; \begin{cases} O\big(m\sqrt{T \ln(d)}\big) & \text{if } m < d \\ O\big(m\sqrt{T \ln(m)}\big) & \text{if } m \geq d. \end{cases}$$

Thus, when $\phi(\mathcal{C}) \subset \mathbb{R}_+^d$, if $\ln |\mathcal{C}| = \omega(\max(\ln(m), \ln(d)))$ and $d = \omega(\ln(m))$, then the LDOMD algorithm with unnormalized negative entropy again gives a better regret bound than both Hedge/Naïve OMD and FPL.

**Case 2:** $\phi(\mathcal{C}) \not\subset \mathbb{R}_+^d$. Here one can no longer use the unnormalized negative entropy in LDOMD. One possibility is to take $\mathcal{K} = \mathbb{R}^d$ and let $F : \mathcal{K} \rightarrow \mathbb{R}$ be defined as $F(x) = \frac{1}{2}\|x\|_2^2$, which leads to an additive update algorithm. In this case, one gets $B_F(\phi(c), x^1) = \frac{1}{2}\|\phi(c) - x^1\|_2^2 \leq 2m^2 \ \forall c \in \mathcal{C}$; moreover, $\|\ell^t\|_2 \leq \sqrt{d} \ \forall t$, and $F$ is 1-strongly convex w.r.t. $\|\cdot\|_2$. Applying Theorem 3 for appropriate $\eta^*$ then gives

$$\mathbf{E}\Big[R_T\big[\,\mathrm{LDOMD}(\eta^*)\,\big]\Big] \; = \; O(m\sqrt{Td}) \,.$$

Thus in general, when $\phi(\mathcal{C}) \not\subset \mathbb{R}_+^d$, LDOMD with squared $L_2$-norm has a similar regret bound as that of Hedge/Naïve OMD and FPL. Note however that in some cases, Hedge/Naïve OMD and FPL may be infeasible to implement efficiently, while LDOMD with squared $L_2$-norm may be efficiently implementable; moreover, in certain cases it may be possible to implement LDOMD with other choices of $\mathcal{K}$ and $F$ that lead to better regret bounds.

In the following sections we will consider several examples of applications of LDOMD to online combinatorial decision problems involving both 0-1 polytopes and general polytopes in $\mathbb{R}^d$.

## 4  Matroid Polytopes

Consider an online decision problem in which the decision space $\mathcal{C}$ contains (not necessarily all) independent sets in a matroid $\mathcal{M} = (E, \mathcal{I})$. Specifically, on each trial $t$, one selects an independent set $I^t \in \mathcal{C}$, and receives a loss vector $\ell^t \in [0, 1]^{|E|}$, with $\ell_e^t$ specifying the loss for including element $e \in E$; the loss for the independent set $I^t$ is given by $\sum_{e \in I^t} \ell_e^t$. Here it is natural to define a mapping $\phi : \mathcal{C} \rightarrow \{0, 1\}^{|E|}$ that maps each independent set $I \in \mathcal{C}$ to its characteristic vector, defined as $\phi_e(I) = \mathbf{1}(e \in I)$; the loss on selecting $I^t \in \mathcal{C}$ is then $\phi(I^t) \cdot \ell^t$. Thus here $d = |E|$, and $\phi(\mathcal{C}) \subseteq \{0, 1\}^{|E|}$. A particularly interesting case is obtained by taking $\mathcal{C}$ to contain all the *maximal* independent sets (bases) in $\mathcal{I}$; in this case, the polytope $\mathrm{conv}(\phi(\mathcal{C}))$ is known as the *matroid base polytope* of $\mathcal{M}$. This polytope, often denoted as $\mathcal{B}(\mathcal{M})$, is also given by

$$\mathcal{B}(\mathcal{M}) = \Big\{ x \in \mathbb{R}^{|E|} \ \Big| \ \textstyle\sum_{e \in S} x_e \leq \mathrm{rank}_{\mathcal{M}}(S) \ \forall S \subset E, \text{ and } \sum_{e \in E} x_e = \mathrm{rank}_{\mathcal{M}}(E) \Big\} ,$$

where $\mathrm{rank}_{\mathcal{M}} : 2^E \rightarrow \mathbb{R}$ is the *matroid rank function* of $\mathcal{M}$ defined as

$$\mathrm{rank}_{\mathcal{M}}(S) = \max \big\{ |I| \mid I \in \mathcal{I}, I \subseteq S \big\} \quad \forall S \subseteq E \,.$$

We will see below (Section 5) that both decomposition and unnormalized relative entropy projection steps in this case can be performed efficiently assuming an appropriate oracle.

We note that Example 1 ($m$-subsets of a ground set of $d$ elements) can be viewed as a special case of the above setting for the matroid $\mathcal{M}_{\mathrm{sub}} = (E, \mathcal{I})$ defined by $E = [d]$ and $\mathcal{I} = \{S \subseteq E \mid |S| \leq m\}$; the set $\mathcal{C}$ of $m$-subsets of $[d]$ is then simply the set of bases in $\mathcal{I}$, and $\mathrm{conv}(\phi(\mathcal{C})) = \mathcal{B}(\mathcal{M}_{\mathrm{sub}})$. The following is another well-studied example:

**Example 2** (Spanning trees with edge-based losses). *Here one is given a connected, undirected graph $G = ([n], E)$, and the decision space $\mathcal{C}$ is the set of all spanning trees in $G$. On each trial $t$, one selects a spanning tree $\mathcal{T}^t \in \mathcal{C}$ and receives a loss vector $\ell^t \in [0, 1]^{|E|}$, with $\ell_e^t$ specifying the loss for using edge $e$; the loss for the tree $\mathcal{T}^t$ is given by $\sum_{e \in \mathcal{T}^t} \ell_e^t$. It is well known that the set of all spanning trees in $G$ is the set of bases in the graphic matroid $\mathcal{M}_G = (E, \mathcal{I})$, where $\mathcal{I}$ contains edge sets of all acyclic subgraphs of $G$. Therefore here $d = |E|$, $\phi(\mathcal{C})$ is the set of incidence vectors of all spanning trees in $G$, and $\mathrm{conv}(\phi(\mathcal{C})) = \mathcal{B}(\mathcal{M}_G)$, also known as the* spanning tree *polytope.*

*Here LDOMD with unnormalized negative entropy has a regret bound of $O\big(n\sqrt{T \ln(\frac{|E|}{n-1})}\big)$.*

## 5  Polytopes Associated with Submodular Functions

Next we consider settings where the decision space $\mathcal{C}$ is in one-to-one correspondence with the set of vertices of the base polytope associated with a submodular function, and losses are linear in the corresponding vertex representations of elements in $\mathcal{C}$. This setting was considered recently in [15], and as we shall see, encompasses both of the examples we saw earlier, as well as many others. Let $f : 2^{[n]} \to \mathbb{R}$ be a submodular function with $f(\emptyset) = 0$. The *base polytope* of $f$ is defined as

$$\mathcal{B}(f) = \left\{ x \in \mathbb{R}^n \;\middle|\; \textstyle\sum_{i \in S} x_i \leq f(S) \; \forall S \subset [n], \text{ and } \sum_{i=1}^n x_i = f([n]) \right\}.$$

Let $\phi : \mathcal{C} \to \mathbb{R}^n$ be a bijective mapping from $\mathcal{C}$ to the vertices of $\mathcal{B}(f)$; thus $\mathrm{conv}(\phi(\mathcal{C})) = \mathcal{B}(f)$.

### 5.1  Monotone Submodular Functions

It is known that when $f$ is a *monotone* submodular function (which means $U \subseteq V \implies f(U) \leq f(V)$), then $\mathcal{B}(f) \subseteq \mathbb{R}_+^n$ [4]. Therefore in this case one can take $\mathcal{K} = \mathbb{R}_+^n$ and $F : \mathcal{K} \to \mathbb{R}$ to be the unnormalized negative entropy. Both decomposition and unnormalized relative entropy projection steps can be performed in time $O(n^6 + n^5 Q)$, where $Q$ is the time taken by an oracle that given $S$ returns $f(S)$; for *cardinality-based* submodular functions, for which $f(S) = g(|S|)$ for some $g : [n] \to \mathbb{R}$, these steps can be performed in just $O(n^2)$ time [15].

**Remark on matroid base polytopes and spanning trees.** We note that the matroid rank function of any matroid $\mathcal{M}$ is a monotone submodular function, and that the matroid base polytope $\mathcal{B}(\mathcal{M})$ is the same as $\mathcal{B}(\mathrm{rank}_{\mathcal{M}})$. Therefore Examples 1 and 2 can also be viewed as special cases of the above setting. For the spanning trees of Example 2, the decomposition step of [14] makes use of a linear programming formulation whose exact time complexity is unclear. Instead, one could use the decomposition step associated with the submodular function $\mathrm{rank}_{\mathcal{M}_G}$, which takes $O(|E|^6)$ time.

Matroid polytopes are 0-1 polytopes; the example below illustrates a more general polytope:

**Example 3** (Permutations with a certain position-based loss). *Let $\mathcal{C} = \mathcal{S}_n$, the set of all permutations of $n$ objects: $\mathcal{C} = \{\sigma : [n] \to [n] \,|\, \sigma \text{ is bijective}\}$. On each trial $t$, one selects a permutation $\sigma^t \in \mathcal{C}$ and receives a loss vector $\ell^t \in [0, 1]^n$; the loss of the permutation is given by $\sum_{i=1}^n \ell_i^t (n - \sigma^t(i) + 1)$. This type of loss arises in scheduling applications, where $\ell_i^t$ denotes the time taken to complete the $i$-th job, and the loss of a job schedule (permutation of jobs) is the total waiting time of all jobs (the waiting time of a job is its own completion time plus the sum of completion times of all jobs scheduled before it) [15]. Here it is natural to define a mapping $\phi : \mathcal{C} \to \mathbb{R}_+^n$ that maps $\sigma \in \mathcal{C}$ to $\phi(\sigma) = (n - \sigma(1) + 1, \ldots, n - \sigma(n) + 1)$; the loss on selecting $\sigma^t \in \mathcal{C}$ is then $\phi(\sigma^t) \cdot \ell^t$. Thus here we have $d = n$, and $\phi(\mathcal{C}) = \{(\sigma(1), \ldots, \sigma(n)) \,|\, \sigma \in \mathcal{S}_n\}$. It is known that the $n!$ vectors in $\phi(\mathcal{C})$ are exactly the vertices of the base polytope corresponding to the monotone (cardinality-based) submodular function $f_{perm} : 2^{[n]} \to \mathbb{R}$ defined as $f_{perm}(S) = \sum_{i=1}^{|S|} (n - i + 1)$. Thus $\mathrm{conv}(\phi(\mathcal{C})) = \mathcal{B}(f_{perm})$; this is a well-known polytope called the* permutahedron *[21], and has recently been studied in the context of online learning applications in [18, 15, 1]. Here $\|\phi(\sigma)\|_1 = \frac{n(n+1)}{2} \; \forall \sigma \in \mathcal{C}$, and therefore LDOMD with unnormalized negative entropy has a regret bound of $O\big(n^2\sqrt{T \ln(n)}\big)$. As noted above, decomposition and unnormalized relative entropy projection steps take $O(n^2)$ time.*

### 5.2  General Submodular Functions

In general, when $f$ is non-monotone, $\mathcal{B}(f) \subset \mathbb{R}^n$ can contain vectors with non-negative entries. Here one can use LOMD with the squared $L_2$-norm. The Euclidean projection step can again be performed in time $O(n^6 + n^5 Q)$ in general, where $Q$ is the time taken by an oracle that given $S$ returns $f(S)$, and in $O(n^2)$ time for cardinality-based submodular functions [15].

# 6 Permutation Polytopes

There has been increasing interest in recent years in online decision problems involving rankings or permutations, largely due to their role in applications such as information retrieval, recommender systems, rank aggregation, etc [12, 18, 19, 15, 1, 2]. Here the decision space is $\mathcal{C} = \mathcal{S}_n$, the set of all permutations of $n$ objects:
$$\mathcal{C} = \{\sigma : [n] \to [n] \mid \sigma \text{ is bijective}\}.$$

On each trial $t$, one predicts a permutation $\sigma^t \in \mathcal{C}$ and receives some type of loss. We saw one special type of loss in Example 3; we now consider any loss that can be represented as a linear function of some vector representation of the permutations in $\mathcal{C}$. Specifically, let $d \in \mathbb{Z}_+$, and let $\phi : \mathcal{C} \to \mathbb{R}^d$ be any injective mapping such that on predicting $\sigma^t$, one receives a loss vector $\ell^t \in [0, 1]^d$ and incurs loss $\phi(\sigma^t) \cdot \ell^t$. For any such mapping $\phi$, the polytope $\mathrm{conv}(\phi(\mathcal{C}))$ is called a *permutation polytope* [5].[3] The permutahedron we saw in Example 3 is one example of a permutation polytope; here we consider various other examples. For any such polytope, if one can perform the decomposition and suitable Bregman projection steps efficiently, then one can use the LDOMD algorithm to obtain good regret guarantees with respect to the associated loss.

**Example 4** (Permutations with assignment-based losses). *Here on each trial $t$, one selects a permutation $\sigma^t \in \mathcal{C}$ and receives a loss matrix $\ell^t \in [0, 1]^{n \times n}$, with $\ell_{ij}^t$ specifying the loss for assigning element $i$ to position $j$; the loss for the permutation $\sigma^t$ is given by $\sum_{i=1}^n \ell_{i,\sigma^t(i)}^t$. Here it is natural to define a mapping $\phi : \mathcal{C} \to \{0, 1\}^{n \times n}$ that maps each $\sigma \in \mathcal{C}$ to its associated permutation matrix $P^\sigma \in \{0, 1\}^{n \times n}$, defined as $P_{ij}^\sigma = \mathbf{1}(\sigma(i) = j) \, \forall i, j \in [n]$; the loss incurred on predicting $\sigma^t \in \mathcal{C}$ is then $\sum_{i=1}^n \sum_{j=1}^n \phi_{ij}(\sigma^t) \ell_{ij}^t$. Thus we have here that $d = n^2$, $\phi(\mathcal{C}) = \{P^\sigma \in \{0, 1\}^{n \times n} \mid \sigma \in \mathcal{S}_n\}$, and $\mathrm{conv}(\phi(\mathcal{C}))$ is the well-known* Birkhoff polytope *containing all doubly stochastic matrices in $[0, 1]^{n \times n}$ (also known as the* assignment polytope *or the* perfect matching polytope of the complete bipartite graph $K_{n,n}$). *Here LDOMD with unnormalized negative entropy has a regret bound of $O\left(n\sqrt{T \ln(n)}\right)$. This recovers exactly the PermELearn algorithm used in [12]; see [12] for efficient implementations of the decomposition and unnormalized relative entropy projection steps.*

**Example 5** (Permutations with general position-based losses). *Here on each trial $t$, one selects a permutation $\sigma^t \in \mathcal{C}$ and receives a loss vector $\ell^t \in [0, 1]^n$. There is a weight function $\gamma : [n] \to \mathbb{R}_+$ that weights the loss incurred at each position, such that the loss contributed by element $i$ is $\ell_i^t \gamma(\sigma^t(i))$; the total loss of the permutation $\sigma^t$ is given by $\sum_{i=1}^n \ell_i^t \gamma(\sigma^t(i))$. Note that the particular loss considered in Example 3 (and in [15]) is a special case of such a position-based loss, with weight function $\gamma(i) = (n-i+1)$. Several other position-dependent losses are used in practice; for example, the discounted cumulative gain (DCG) based loss, which is widely used in information retrieval applications, effectively uses $\gamma(i) = 1 - \frac{1}{\log_2(i)+1}$ [9]. For a general position-based loss with weight function $\gamma$, one can define $\phi : \mathcal{C} \to \mathbb{R}_+^n$ as $\phi(\sigma) = (\gamma(\sigma(1)), \ldots, \gamma(\sigma(n)))$. This yields a permutation polytope $\mathrm{conv}(\phi(\mathcal{C})) = \mathrm{conv}\left(\{(\gamma(\sigma(1)), \ldots, \gamma(\sigma(n))) \mid \sigma \in \mathcal{S}_n\}\right) \subset \mathbb{R}_+^n$. Provided one can implement the decomposition and suitable Bregman projection steps efficiently, one can use the LDOMD algorithm to get a sublinear regret.*

# 7 Application to an Online Transportation Problem

Consider now the following transportation problem: there are $m$ supply locations for a particular commodity and $n$ demand locations, with a supply vector $a \in \mathbb{Z}_+^m$ and demand vector $b \in \mathbb{Z}_+^n$ specifying the (integer) quantities of the commodity supplied/demanded by the various locations. Assume $\sum_{i=1}^m a_i = \sum_{j=1}^n b_j \stackrel{\triangle}{=} q$. In the offline setting, there is a cost matrix $\ell \in [0, 1]^{m \times n}$, with $\ell_{ij}$ specifying the cost of transporting one unit of the commodity from supply location $i$ to demand location $j$, and the goal is to decide on a transportation matrix $Q \in \mathbb{Z}_+^{m \times n}$ that specifies suitable (integer) quantities of the commodity to be transported between the various supply and demand locations so as to minimize the total transportation cost, $\sum_{i=1}^m \sum_{j=1}^n Q_{ij} \ell_{ij}$.

Here we consider an online variant of this problem where the supply vector $a$ and demand vector $b$ are viewed as remaining constant over some period of time, while the costs of transporting the com-

**Algorithm** Decomposition Step for Transportation Polytopes

---

**Input:** $X \in \mathcal{T}(a,b)$   (where $a \in \mathbb{Z}_+^m$, $b \in \mathbb{Z}_+^n$)

**Initialize:** $A^1 \leftarrow X$; $k \leftarrow 0$

**Repeat:**
- $k \leftarrow k+1$
- Find an extreme point $Q^k \in \mathcal{T}(a,b)$ such that $A_{ij}^k = 0 \implies Q_{ij}^k = 0$   (see Appendix B)
- $\alpha_k \leftarrow \min_{(i,j):Q_{ij}^k > 0} \left( \frac{A_{ij}^k}{Q_{ij}^k} \right)$
- $A^{k+1} \leftarrow A^k - \alpha_k Q^k$

**Until** ( all entries of $A^{k+1}$ are zero )

**Ouput:** Decomposition of $X$ as convex combination of extreme points $Q^1, \ldots, Q^k$:
$\quad X = \sum_{r=1}^k \alpha_r Q^r$   (it can be verified that $\alpha_r \in (0,1] \; \forall r$ and $\sum_{r=1}^k \alpha_r = 1$)

---

Figure 3: Decomposition step in applying LDOMD to transportation polytopes.

modity between various supply and demand locations change over time. Specifically, the decision space here is the set of all valid (integer) transportation matrices satisfying constraints given by $a, b$:

$$\mathcal{C} = \left\{ Q \in \mathbb{Z}_+^{m \times n} \mid \sum_{j=1}^n Q_{ij} = a_i \; \forall i \in [m], \; \sum_{i=1}^m Q_{ij} = b_j \; \forall j \in [n] \right\}.$$

On each trial $t$, one selects a transportation matrix $Q^t \in \mathcal{C}$, and receives a cost matrix $\ell^t \in [0,1]^{m \times n}$; the loss incurred is $\sum_{i=1}^m \sum_{j=1}^n Q_{ij}^t \ell_{ij}^t$. A natural mapping here is simply the identity: $\phi : \mathcal{C} \rightarrow \mathbb{Z}_+^{m \times n}$ with $\phi(Q) = Q \; \forall Q \in \mathcal{C}$. Thus we have here $d = mn$, $\phi(\mathcal{C}) = \mathcal{C}$, and $\mathrm{conv}(\phi(\mathcal{C}))$ is the well-known *transportation polytope* $\mathcal{T}(a,b)$ (e.g. see [6]):

$$\mathrm{conv}(\phi(\mathcal{C})) = \mathcal{T}(a,b) = \left\{ X \in \mathbb{R}_+^{m \times n} \mid \sum_{j=1}^n X_{ij} = a_i \; \forall i \in [m], \; \sum_{i=1}^m X_{ij} = b_j \; \forall j \in [n] \right\}.$$

Transportation polytopes generalize the Birkhoff polytope of doubly stochastic matrices, which can be seen to arise as a special case when $m = n$ and $a_i = b_i = 1 \; \forall i \in [n]$ (see Example 4). While the Birkhoff polytope is a 0-1 polytope, a general transportation polytope clearly includes non-Boolean vertices. Nevertheless, we do have $\mathcal{T}(a,b) \subset \mathbb{R}_+^{m \times n}$, which suggests we can use the LDOMD algorithm with unnormalized negative entropy.

For the decomposition step in LDOMD, one can use an algorithm broadly similar to that used for the Birkhoff polytope in [12]. Specifically, given a matrix $X \in \mathrm{conv}(\phi(\mathcal{C})) = \mathcal{T}(a,b)$, one successively subtracts off multiples of extreme points $Q^k \in \mathcal{C}$ from $X$ until one is left with a zero matrix (see Figure 3). However, a key step of this algorithm is to find a suitable extreme point to subtract off on each iteration. In the case of the Birkhoff polytope, this involved finding a suitable permutation matrix, and was achieved by finding a perfect matching in a suitable bipartite graph. For general transportation polytopes, we make use of a characterization of extreme points in terms of spanning forests in a suitable bipartite graph (see Appendix B for details). The overall decomposition results in a convex combination of at most $mn$ extreme points in $\mathcal{C}$, and takes $O(m^3 n^3)$ time.

The unnormalized relative entropy projection step can be performed efficiently by using a procedure similar to the Sinkhorn balancing used for the Birkhoff polytope in [12]. Specifically, given a non-negative matrix $\widetilde{X} \in \mathbb{R}_+^{m \times n}$, one alternately scales the rows and columns to match the desired row and column sums until some convergence criterion is met. As with Sinkhorn balancing, this results in an approximate projection step, but does not hurt the overall regret analysis (other than a constant additive term), yielding a regret bound of $O\big(q\sqrt{T \ln(\max(mn, q))}\big)$.

## 8  Conclusion

We have considered a general form of online combinatorial decision problems, where costs can be linear in any suitable low-dimensional vector representation of elements of the decision space, and have given a general algorithm termed *low-dimensional online mirror descent* (LDOMD) for such problems. Our study emphasizes the role of the convex polytope arising from the vector representation of the decision space; this both yields a unification and generalization of previous algorithms, and gives a general framework that can be used to design new algorithms for specific applications.

**Acknowledgments.** Thanks to the anonymous reviewers for helpful comments and Chandrashekar Lakshminarayanan for helpful discussions. AR is supported by a Microsoft Research India PhD Fellowship. SA thanks DST and the Indo-US Science & Technology Forum for their support.

## Footnotes

[1]We note that the recent online stochastic mirror descent (OSMD) algorithm of Audibert et al. [3] also generalizes the Component Hedge algorithm, but in a different direction: OSMD (as described in [3]) applies to only Boolean representations, but allows also for partial information (bandit) settings; here we consider only full information settings, but allow for more general vector representations.

[2]Recall that for a closed convex set $\mathcal{K} \subseteq \mathbb{R}^d$, a function $F : \mathcal{K} \to \mathbb{R}$ is Legendre if it is strictly convex, differentiable on $\mathrm{int}(\mathcal{K})$, and (for any norm $\|\cdot\|$ on $\mathbb{R}^d$) $\|\nabla F(x_n)\| \to +\infty$ whenever $\{x_n\}$ converges to a point in the boundary of $\mathcal{K}$.

[3]The term 'permutation polytope' is sometimes used to refer to various polytopes obtained through specific mappings $\phi : \mathcal{S}_n \to \mathbb{R}^d$; here we use the term in a broad sense for any such polytope, following terminology of Bowman [5]. (Note that the description Bowman [5] gives of a particular 0-1 permutation polytope in $\mathbb{R}^{n(n-1)}$, known as the binary choice polytope or the linear ordering polytope [20], is actually incorrect; e.g. see [11].)

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
