[Supplementary Material]

# Online Decision-Making in General Combinatorial Spaces

## A  Supplement to Section 2 (Preliminaries and Background)

### A.1  Online Mirror Descent (OMD) for Online Linear Optimization

---

**Algorithm** Online Mirror Descent (OMD) for Online Linear Optimization

---

**Inputs:**
    Convex set $\Omega \subseteq \mathbb{R}^n$

**Parameters:**
    $\eta > 0$
    Closed convex set $\mathcal{K} \supseteq \Omega$, Legendre function $F : \mathcal{K} \to \mathbb{R}$

**Initialize:**
    $x^1 \in \operatorname{argmin}_{x \in \Omega} F(x)$  (or $x^1 =$ any other point in $\Omega$)

**For** $t = 1 \ldots T$:
  – Receive loss vector $\ell^t \in \mathbb{R}^n$
  – Incur loss $x^t \cdot \ell^t$
  – Update:
    $\widetilde{x}^{t+1} \leftarrow \nabla F^*(\nabla F(x^t) - \eta \ell^t)$
    $x^{t+1} \leftarrow \operatorname{argmin}_{x \in \Omega} B_F(x, \widetilde{x}^{t+1})$

---

The following bound on the regret of OMD (in the linear setting) is well known (e.g. see [7]):

**Theorem 4** (Regret bound for OMD). *Let $B_F(x, x^1) \leq D^2 \ \forall x \in \Omega$. Let $\| \cdot \|$ be any norm in $\mathbb{R}^n$ such that $\|\ell^t\| \leq G \ \forall t \in [T]$, and such that the restriction of $F$ to $\Omega$ is $\alpha$-strongly convex w.r.t. $\| \cdot \|_*$, the dual norm of $\| \cdot \|$. Then setting $\eta^* = \frac{D}{G}\sqrt{\frac{2\alpha}{T}}$ gives*

$$R_T\big[\, \text{OMD}(\eta^*)\big] \ \Big( = \ \textstyle\sum_{t=1}^T x^t \cdot \ell^t - \inf_{x \in \Omega} \sum_{t=1}^T x \cdot \ell^t \Big) \ \leq \ DG\sqrt{\tfrac{2T}{\alpha}}\,.$$

### A.2  Hedge/Naïve OMD for Online Combinatorial Decision-Making

---

**Algorithm** Hedge/Naïve OMD for Online Combinatorial Decision-Making [10]

---

**Inputs:**
    Finite set of combinatorial structures $\mathcal{C}$
    Mapping $\phi : \mathcal{C} \to \mathbb{R}^d$

**Parameters:**
    $\eta > 0$
**Initialize:**
    $p^1 = \big(\frac{1}{|\mathcal{C}|}, \ldots, \frac{1}{|\mathcal{C}|}\big) \in \Delta_{\mathcal{C}}$
**For** $t = 1 \ldots T$:
  – Randomly draw $c^t \sim p^t$
  – Receive loss vector $\ell^t \in [0,1]^d$
  – Incur loss $\phi(c^t) \cdot \ell^t$
  – Update:
    $\forall c \in \mathcal{C} : \ p_c^{t+1} \leftarrow \dfrac{p_c^t \, \exp(-\eta \, \phi(c) \cdot \ell^t)}{Z^t}\,,$
      where $Z^t = \sum_{c' \in \mathcal{C}} p_{c'}^t \, \exp(-\eta \, \phi(c') \cdot \ell^t)$

---

### A.3 Follow the Perturbed Leader (FPL) for Online Combinatorial Decision-Making

---

**Algorithm** Follow the Perturbed Leader (FPL) for Online Combinatorial Decision-Making [13]

---

**Inputs:**
   Finite set of combinatorial structures $\mathcal{C}$
   Mapping $\phi : \mathcal{C} \to \mathbb{R}^d$

**Parameters:**
   $\eta > 0$
**For** $t = 1 \ldots T$:
   – Draw $z^t \in \left[0, \frac{1}{\eta}\right]^d$ uniformly at random
   – Predict $c^t \in \operatorname{argmin}_{c \in \mathcal{C}} \; \phi(c) \cdot \left( \sum_{s=1}^{t-1} \ell^s + z^t \right)$
   – Receive loss vector $\ell^t \in [0, 1]^d$
   – Incur loss $\phi(c^t) \cdot \ell^t$

---

## B  Supplement to Section 7 (Transportation Polytopes)

The decomposition step in applying LDOMD to transportation polytopes requires finding a suitable extreme point on each iteration. Here we give details of how one can find such an extreme point.

We start by giving a procedure which, given a matrix $X \in \mathcal{T}(a, b)$, efficiently finds an extreme point $Q \in \mathcal{T}(a, b)$ such that $X_{ij} = 0 \implies Q_{ij} = 0$ (note that such an extreme point always exists, since $X$ can be written as a convex combination of extreme points, all of which must necessarily have a zero entry wherever $X$ does). We will make use of the following characterization of extreme points of transportation polytopes in terms of spanning forests of complete bipartite graphs (e.g. see [6]):

**Theorem 5** (Characterization of extreme points of transportation polytopes)**.** *Let $a \in \mathbb{Z}_+^m$, $b \in \mathbb{Z}_+^n$. A matrix $X \in \mathcal{T}(a, b)$ is an extreme point of $\mathcal{T}(a, b)$ if and only if the edges $\{(i, j) : X_{ij} > 0\}$ form a spanning forest of the complete bipartite graph $K_{m,n}$.*

The basic idea behind the procedure below is as follows: given $X \in \mathcal{T}(a, b)$, let $E = \{(i, j) : X_{ij} > 0\}$. If $E$ forms a spanning forest of $K_{m,n}$, then by Lemma 5, $X$ is already an extreme point. Otherwise, successively remove cycles from $E$ and adjust corresponding entries in $X$ so that $X$ remains in $\mathcal{T}(a, b)$ while satisfying $X_{ij} > 0 \iff (i, j) \in E$. Eventually, $E$ must be a spanning forest of $K_{m,n}$, and therefore by Lemma 5, $X$ must be an extreme point of $\mathcal{T}(a, b)$.

---

**Algorithm** Procedure for finding an extreme point $Q$ of $\mathcal{T}(a, b)$ such that
$\qquad X_{ij} = 0 \implies Q_{ij} = 0$ for a given matrix $X \in \mathcal{T}(a, b)$

---

**Input:**
   $X \in \mathcal{T}(a, b)$   (where $a \in \mathbb{Z}_+^m$, $b \in \mathbb{Z}_+^n$)

**Initialize:**
   $E \leftarrow \{(i, j) : X_{ij} > 0\}$
**While** $\big($ $E$ does not form a spanning forest of $K_{m,n}$ $\big)$ **do**:
   – Find a cycle $E' = \{(i_1, j_1), (i_2, j_1), (i_2, j_2), \ldots, (i_s, j_s), (i_{s+1} = i_1, j_s)\} \subseteq E$ for some $s \geq 2$
   – Let $e_{\min} \in \operatorname{argmin}_{e \in E'} X_e$
   – $\theta \leftarrow \begin{cases} +1 & \text{if } e_{\min} = (i_r, j_r) \text{ for some } r \in [s] \\ -1 & \text{if } e_{\min} = (i_{r+1}, j_r) \text{ for some } r \in [s] \end{cases}$
   – **For** $r = 1 \ldots s$ **do**:
      $X_{i_r, j_r} \leftarrow X_{i_r, j_r} - \theta X_{e_{\min}}$
      $X_{i_{r+1}, j_r} \leftarrow X_{i_{r+1}, j_r} + \theta X_{e_{\min}}$
   – $E \leftarrow \{(i, j) : X_{ij} > 0\}$
**end while**
$Q \leftarrow X$
**Output:** $Q$

---

**Applying the above procedure to implement decomposition step.** The above procedure can be used to implement the decomposition step for transportation polytopes in Section 7 by doing the following on each iteration $k$:

- Apply the above procedure to the matrix $A^k$, which can be verified to belong to $\mathcal{T}(\gamma_k a, \gamma_k b)$ for suitable $\gamma_k \in \mathbb{R}_+$ (specifically, $\gamma_k = 1 - \sum_{r=1}^{k-1} \alpha_r$), to get an extreme point $\widetilde{Q}^k \in \mathcal{T}(\gamma_k a, \gamma_k b)$ satisfying $A_{ij}^k = 0 \implies \widetilde{Q}_{ij}^k = 0$.
- Set $Q^k \leftarrow \frac{1}{\gamma_k} \widetilde{Q}^k$.

It can be verified that $Q^k$ is then an extreme point of $\mathcal{T}(a, b)$ and satisfies $A_{ij}^k = 0 \implies Q_{ij}^k = 0$ as desired.