[Reviews · NeurIPS 2014]

Submitted by Assigned_Reviewer_5

This paper proposes an algorithm for online combinatorial optimization. In this online learning problem, the action space is combinatorially large and can be represented in a d-dimensional Euclidean space such that the loss in each time step is a linear function of the action.

It would greatly improve the paper if there was a thorough comparison between the new algorithm and Online Stochastic Mirror Descent (OSMD by Audibert et al., [3] in the current paper) both in terms of how the algorithms work and in terms of regret bounds. In the current form of the paper, I am not sure if the new algorithm is significantly different from OSMD or if it improves its bounds.
Summary: I can not see the significance of this paper over existing Online Combinatorial Optimization works.

**Update**:
The paper is an improvement over existing work in that it works not only for 0/1-valued vectors, but any Euclidean subspace. I updated my score.

Submitted by Assigned_Reviewer_33

The paper considers online learning of combinatorial decision problems.
A new method LDOMD is introduced, which generalizes the Component Hedge
algorithm of Koolen et al. [14] for 0/1-polytopes and the method of
Suehiro et al. [15] for polytopes associated with submodular functions.
In Theorem 3, a general regret bound is established for LDOMD, which is
claimed to be optimal for 0/1-polytopes. Some new polytopes are
considered, but, with the possible exception of the transportation
polytope, they appear to be minor generalizations of the polytopes of
[14] and [15]. Thus, the main contribution of the paper appears to be a
unification of the results of [14] and [15].

*** Changes after author response ***

[Deleted claim that Thm 3 is suboptimal, which was incorrect, as the
authors pointed out in their response.]

The provided unification is satisfying, but does not seem to lead to
many new applications. The exception might be the transportation
polytope, which the authors argue is "truly novel and non-trivial".

On the whole, I would judge the paper as borderline. I am not sure that
the transportation polytope is technically a very large step forward,
but it does appear practically important. This tips the balance towards
accept.

*** End of changes after author response ***

Other remarks:

Since both [14] and [15] give bounds in terms of L* instead of T, it
should be discussed that the weakening to T is still sufficiently
interesting.

Sections 3.1,3.2: It seems unfair to apply Hedge to the whole class cal{C}
instead of only the subset with ||phi(c)||_1 <= m, which would
give a better bound that would be easier to compare to the
bound for LDOMD.

line 196: although true, it is actually not so easy to see that F is
1/m-strongly convex, and, since x^1 is not explicitly
specified, it is also not that obvious that B_F(phi(c),x^1) <= m log(d/m).

Summary: The paper unifies the results of [14] and [15]. This does not seem to lead to many new applications, except for the transportation polytope. Although I am not sure that the transportation polytope is technically a very large step forward, it does appear practically important.

Submitted by Assigned_Reviewer_41

The paper considers the Online Mirror Descent (OMD) algorithm applied to combinatorial online learning, where the decision space is a set C of combinatorial structures and the loss is linear in a vector representation of the structures. The authors propose a generalization of OMD parameterized by the choice of the mapping phi from the structures in C to their vector representations. The general regret bound follows straightforwardly from the standard OMD analysis. The bulk of the paper focuses on showing that several online combinatorial learning problems studied in the past are special cases of this generalized setting. The analysis of the regret for these special cases boils down to studying the properties of the polytope resulting by taking the convex hull of the image of C according to phi.

The paper is well written and technically sound.

The main strength of this work is the analysis of a variety of problems based on the common unifying language of polytope representation, using the appropriate mathematical tools. On the other hand, there are not a lot of new ideas, especially from the algorithmic viewpoint. The only truly novel application, transportation polytopes, is somewhat disappointing because it uses a trivial mapping phi (the identity function).

ADDED AFTER REBUTTAL
Thanks for clarifying the issue concerning the transportation example. I now understand better the paper's contribution. The score has been adjusted accordingly.
Summary: The paper is interesting because it provides a nice and general parameterization, through the phi mapping, of the well-known OMD algorithm for combinatorial online learning. However, besides encompassing several known results as special cases, it fails to provide a truly convincing novel application.
Author Feedback
Author rebuttal: Thanks to all the reviewers for their comments. As we explain below, Reviewer 1's main concern that "[the paper's] main result is significantly weaker than the result of [14]" is actually not correct; our result does in fact yield an improvement of \sqrt{m} over other algorithms (and therefore, our result is NOT weaker than that of [14]). Reviewer 2's comment on the novel application to transportation polytopes involving a trivial identity mapping misses the point that this is only because we have chosen to describe the elements of the combinatorial space already as transportation matrices for ease of exposition; had we chosen to represent them as weighted graphs, then the mapping would clearly be "non-trivial" (note that the same observation applies to many previously studied decision problems including subsets, spanning trees, permutation matrices, etc). We believe the resulting online learning algorithm we have given for transportation polytopes, which generalizes the permutation learning algorithm of Helmbold & Warmuth (2009), is truly a non-trivial and interesting contribution in its own right. Finally, Reviewer 3 claims that our paper does not contribute anything significant beyond the OSMD algorithm of Audibert et al (2013); as we explain below, OSMD focuses on settings where decisions are represented as Boolean vectors, whereas our work considers more general vector representations in R^d and emphasizes the role of the resulting convex polytopes, which leads to both a unified view of previous work and novel applications such as the new online learning algorithm we have described for transportation polytopes in Section 7. We believe these are significant contributions and urge the area chair and reviewers to re-consider their comments in light of the detailed responses below. Thanks in advance.

Reviewer 1 (A_R_33)

Section 3.1: The regret bound for Hedge here should be O(m \sqrt{T m \ln(d/m)). Thanks for catching this; we will certainly fix this. Please note however that the regret bound we have given for LDOMD is indeed optimal, and is comparable to that of [14] for 0/1 polytopes. In particular, the bound for Component Hedge (CH) in [14], which involves the loss L^* of the best single decision in hindsight, is of the form O(\sqrt{L^* m \ln(d/m)} + m \ln(d/m)). However, in the worst case, L^* = O(T m), in which case this bound becomes O(m \sqrt{T \ln(d/m)}). This is the same as the bound we get for LDOMD in the 0/1 polytope setting. Just as the bound for CH involving L^* improves on the bound for Hedge involving L^* by a factor of \sqrt{m} (see [14]), similarly, in our case too, the bound for LDOMD in terms of T improves on the above bound for Hedge in terms of T by a factor of \sqrt{m}. Therefore, our result is NOT weaker than that of [14].

Giving bounds in terms of L^*: We gave worst-case bounds in terms of T since bounds in term of L^* require knowledge of L^* (or a suitable upper bound on L^*) in order to tune the parameter \eta, but we can certainly include bounds in terms of L^* as well. The relative improvements over Hedge and FPL algorithms still hold.

Reviewer 2 (A_R_41)

Application to transportation polytopes: Please note that we chose to represent the combinatorial decision space here in terms of transportation matrices only for simplicity of exposition, and that the mapping \phi being the identity function is just an artifact of this choice (we could just as well have represented decisions as transportation graphs, in which case the mapping would have been at least as non-trivial as that for spanning trees or permutation matrices). The resulting online learning algorithm we have given for transportation polytopes generalizes the online permutation learning algorithm of Helmbold & Warmuth (2009) and involves novel routines for the associated decomposition and projection steps. We believe this is a truly novel and non-trivial example of how the unified view offered by our emphasis on polytopes arising from general vector representations can lead to new applications and online learning algorithms.

Reviewer 3 (A_R_5)

LDOMD vs. OSMD and contribution of the paper: The OSMD algorithm of Audibert et al, 2013 [3] is designed for both partial information (bandit) settings and full information settings. The LDOMD algorithm we describe focuses on only full-information settings. However, as the OSMD paper [3] clearly states in several places, the focus there is on settings where elements of the decision space are represented as Boolean vectors:

- [3], abstract, 1st line: "We address online linear optimization problems when the possible actions of the decision maker are represented by binary vectors."
- [3], p.2: "we assume that the action set A is a subset of the d-dimensional hypercube {0,1}^d …"
- [3], p.8: "In this paper we restrict our attention to the combinatorial learning setting in which A is a subset of {0,1}^d and the loss is linear."

The LDOMD algorithm we have described and its analysis are for general vector representations in R^d. Of course, both algorithms are very similar as both are derived from the basic online mirror descent (OMD) algorithm for online linear optimization. We would be happy to specifically point this out.

We would also like to point out that we do not view the LDOMD algorithm itself (or its regret analysis) as the primary contribution of our work, but rather the unification and generalization it offers of previous work, including the Component Hedge and OSMD algorithms that have focused on decisions represented as Boolean vectors, and the recent work of Suehiro et al where decisions are represented as vertices of submodular polytopes. In addition, we believe that our emphasis on the role of the convex polytopes arising from various vector representations is a useful viewpoint that can lead to new applications, such as the new application to transportation polytopes we have described in Section 7.